# Max-Margin Invariant Features from Transformed Unlabeled Data

**Dipan K. Pal, Ashwin A. Kannan,** *Gautam Arakalgud,* **Marios Savvides**
Department of Electrical and Computer Engineering
Carnegie Mellon University
Pittsburgh, PA 15213
{dipanp,aalapakk,garakalgud,marioss}@cmu.edu

## Abstract

The study of representations invariant to common transformations of the data is important to learning. Most techniques have focused on local approximate invariance implemented within expensive optimization frameworks lacking explicit theoretical guarantees. In this paper, we study kernels that are invariant to a unitary group while having theoretical guarantees in addressing the important practical issue of unavailability of transformed versions of *labelled* data. A problem we call the *Unlabeled Transformation Problem* which is a special form of semi-supervised learning and one-shot learning. We present a theoretically motivated alternate approach to the invariant kernel SVM based on which we propose Max-Margin Invariant Features (MMIF) to solve this problem. As an illustration, we design an framework for face recognition and demonstrate the efficacy of our approach on a large scale semi-synthetic dataset with 153,000 images and a new challenging protocol on Labelled Faces in the Wild (LFW) while out-performing strong baselines.

## 1 Introduction

It is becoming increasingly important to learn well generalizing representations that are invariant to many common nuisance transformations of the data. Indeed, being *invariant* to intra-class transformations while being *discriminative* to between-class transformations can be said to be one of the fundamental problems in pattern recognition. The nuisance transformations can give rise to many 'degrees of freedom' even in a constrained task such as face recognition (*e.g.* pose, age-variation, illumination etc.). Explicitly factoring them out leads to improvements in recognition performance as found in [10, 7, 6]. It has also been shown that that features that are explicitly invariant to intra-class transformations allow the sample complexity of the recognition problem to be reduced [2]. To this end, the study of invariant representations and machinery built on the concept of explicit invariance is important.

**Invariance through Data Augmentation.** Many approaches in the past have enforced invariance by generating transformed *labelled* training samples in some form such as [13, 17, 19, 9, 15, 4]. Perhaps, one of the most popular method for incorporating invariances in SVMs is the virtual support method (VSV) in [18], which used sequential runs of SVMs in order to find and augment the support vectors with transformed versions of themselves.

**Indecipherable transformations in data leads to shortage of transformed labelled samples.** The above approaches however, assume that one has *explicit* knowledge about the transformation. This is a strong assumption. Indeed, in most general machine learning applications, the transformation

---

present in the data is not clear and cannot be modelled easily, *e.g.* transformations between different views of a general 3D object and between different sentences articulated by the same person. Methods which work on generating invariance by *explicitly* transforming or augmenting *labelled* training data cannot be applied to these scenarios. Further, in cases where we do know the transformations that exist and we actually can model them, it is difficult to generate transformed versions of very large labelled datasets. Hence there arises an important problem: how do we train models to be invariant to transformations in *test* data, when we *do not* have access to transformed *labelled* training samples ?

**Availability of unlabeled transformed data.**
Although it is difficult to obtain or generate transformed labelled data (due to the reasons mentioned above), *unlabeled* transformed data is more readily available. For instance, if different views of *specific* objects of interest are not available, one can simply collect views of general objects. Also, if different sentences spoken by a *specific* group of people are not available, one can simply collect those spoken by members of the general population. In both these scenarios, no *explicit* knowledge or model of the transformation is needed, thereby bypassing the problem of indecipherable transformations. This situation is common in vision *e.g.* only unlabeled transformed images are observed, but has so far mostly been addressed by the community by intense efforts in large scale data collection. Note that the transformed data that is collected is *not* required to be labelled. We now are in a position to state the central problem that this paper addresses.

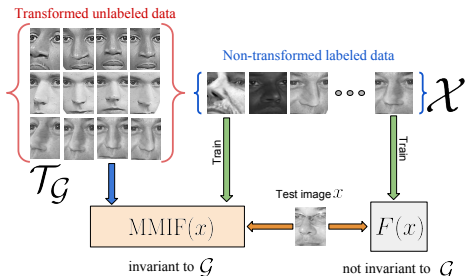

**Figure 1:** Max-Margin Invariant Features (MMIF) can solve an important problem we call the Unlabeled Transformation Problem. In the figure, a traditional classifier $F(x)$ "learns" invariance to nuisance transformations directly from the labeled dataset $\mathcal{X}$. On the other hand, our approach (MMIF) can incorporate additional invariance learned from any unlabeled data that undergoes the nuisance transformation of interest.

**The Unlabeled Transformation (UT) Problem:**
*Having access to transformed versions of the training unlabeled data but not of labelled data, how do we learn a discriminative model of the labelled data, while being invariant to transformations present in the unlabeled data ?*

**Overall approach.** The approach presented in this paper however (see Fig. 1), can solve this problem and learn invariance to transformations observed only through unlabeled samples and does not need *labelled* training data augmentation. We explicitly and simultaneously address both problems of generating invariance to intra-class transformation (through invariant kernels) and being discriminative to inter or between class transformations (through max-margin classifiers). Given a new test sample, the final extracted feature is invariant to the transformations observed in the unlabeled set, and thereby generalizes using just a single example. This is an example of one-shot learning.

**Prior Art: Invariant Kernels.** Kernel methods in machine learning have long been studied to considerable depth. Nonetheless, the study of *invariant kernels* and techniques to extract invariant features has received much less attention. An invariant kernel allows the kernel product to remain invariant under transformations of the inputs. Most instances of incorporating invariances focused on *local* invariances through regularization and optimization such as [18, 19, 3, 21]. Some other techniques were jittering kernels [17, 3] and tangent-distance kernels [5], both of which sacrificed the positive semi-definite property of its kernels and were computationally expensive. Though these methods have had some success, most of them still lack explicit theoretical guarantees towards invariance. The proposed invariant kernel SVM formulation on the other hand, develops a valid PSD kernel that is guaranteed to be invariant. [4] used group integration to arrive at invariant kernels but did not address the Unlabeled Transformation problem which our proposed kernels do address. Further, our proposed kernels allow for the formulation of the invariant SVM and application to large scale problems. Recently, [14] presented some work with invariant kernels. However, unlike our non-parametric formulation, they do not learn the group transformations from the data itself and assume known parametric transformations (*i.e.* they assume that transformation is computable).

**Key ideas.** The key ideas in this paper are twofold.

1. The first is to model transformations using *unitary groups* (or sub-groups) leading to unitary-group invariant kernels. Unitary transforms allow the dot product to be preserved and allow for interesting generalization properties leading to low sample complexity and also allow learning transformation invariance from unlabeled examples (thereby solving the Unlabeled Transformation Problem). Classes of learning problems, such as vision, often have transformations belonging to a unitary-group, that one would like to be invariant towards (such as translation and rotation). In practice however, [8] found that invariance to much more general transformations not captured by this model can been achieved.

2. Secondly, we combine max-margin classifiers with invariant kernels leading to non-linear max-margin unitary-group invariant classifiers. These theoretically motivated invariant non-linear SVMs form the foundation upon which Max-Margin Invariant Features (MMIF) are based. MMIF features can effectively solve the important Unlabeled Transformation Problem. To the best of our knowledge, this is the first theoretically proven formulation of this nature.

**Contributions.** In contrast to many previous studies on invariant kernels, we study non-linear positive semi-definite unitary-group invariant kernels guaranteeing invariance that can address the UT Problem. One of our central theoretical results to applies group integration in the RKHS. It builds on the observation that, under unitary restrictions on the kernel map, group action in the input space is reciprocated in the RKHS. Using the proposed invariant kernel, we present a theoretically motivated approach towards a non-linear invariant SVM that can solve the UT Problem with explicit invariance guarantees. As our main theoretical contribution, we showcase a result on the generalization of max-margin classifiers in group-invariant subspaces. We propose Max-Margin Invariant Features (MMIF) to learn highly discriminative non-linear features that also solve the UT problem. On the practical side, we propose an approach to face recognition to combine MMIFs with a pre-trained deep learning feature extractor (in our case VGG-Face [12]). MMIF features can be used with deep learning whenever there is a need to focus on a particular transformation in data (in our application pose in face recognition) and can further improve performance.

## 2  Unitary-Group Invariant Kernels

**Premise:** Consider a dataset of normalized samples along with labels $\mathcal{X} = \{x_i\}, \mathcal{Y} = \{y_i\} \; \forall i \in 1...N$ with $x \in \mathbb{R}^d$ and $y \in \{+1, -1\}$. We now introduce into the dataset a number of unitary transformations $g$ part of a locally compact unitary-group $\mathcal{G}$. We note again that the set of transformations under consideration need not be the entire unitary group. They could very well be a subgroup. Our augmented normalized dataset becomes $\{gx_i, y_i\} \; \forall g \in \mathcal{G} \; \forall i$. For clarity, we denote by $gx$ the action of group element $g \in \mathcal{G}$ on $x$, *i.e.* $gx = g(x)$. We also define an *orbit* of $x$ under $\mathcal{G}$ as the set $\mathcal{X}_\mathcal{G} = \{gx\} \; \forall g \in \mathcal{G}$. Clearly, $\mathcal{X} \subseteq \mathcal{X}_\mathcal{G}$. An invariant function is defined as follows.

**Definition 2.1** (*$\mathcal{G}$-Invariant Function*). For any group $\mathcal{G}$, we define a function $f : \mathcal{X} \to \mathbb{R}^n$ to be $\mathcal{G}$-invariant if $f(x) = f(gx) \; \forall x \in \mathcal{X} \; \forall g \in \mathcal{G}$.

One method of generating an invariant towards a group is through group integration. Group integration has stemmed from classical invariant theory and can be shown to be a projection onto a $\mathcal{G}$-invariant subspace for vector spaces. In such a space $x = gx \; \forall g \in \mathcal{G}$ and thus the representation $x$ is invariant under the transformation of any element from the group $\mathcal{G}$. This is ideal for recognition problems where one would want to be discriminative to between-class transformations (for *e.g.* between distinct subjects in face recognition) but be *invariant to within-class transformations* (for *e.g.* different images of the same subject). The set of transformations we model as $\mathcal{G}$ are the within-class transformations that we would like to be invariant towards. An invariant to any group $\mathcal{G}$ can be generated through the following basic (previously) known property (Lemma 2.1) based on group integration.

**Lemma 2.1.** *(Invariance Property) Given a vector $\omega \in \mathbb{R}^d$, and any affine group $\mathcal{G}$, for any fixed $g' \in \mathcal{G}$ and a normalized Haar measure $dg$, we have $g' \int_\mathcal{G} g\omega \, dg = \int_\mathcal{G} g\omega \, dg$*

The Haar measure $(dg)$ exists for every locally compact group and is unique up to a positive multiplicative constant (hence normalized). A similar property holds for discrete groups. Lemma 2.1 results in the quantity $\int_\mathcal{G} g\omega \, dg$ enjoy global invariance (encompassing all elements) to group $\mathcal{G}$.

This property allows one to generate a $\mathcal{G}$-invariant subspace in the inherent space $\mathbb{R}^d$ through group integration. In practice, the integral corresponds to a summation over transformed samples. The

following two lemmas (novel results, and part of our contribution) (Lemma 2.2 and 2.3) showcase elementary properties of the operator $\Psi = \int_{\mathcal{G}} g \, dg$ for a unitary-group $\mathcal{G}$ [2]. These properties would prove useful in the analysis of unitary-group invariant kernels and features.

**Lemma 2.2.** *If $\Psi = \int_{\mathcal{G}} g \, dg$ for unitary $\mathcal{G}$, then $\Psi^T = \Psi$*

**Lemma 2.3.** *(Unitary Projection) If $\Psi = \int_{\mathcal{G}} g \, dg$ for any affine $\mathcal{G}$, then $\Psi\Psi = \Psi$, i.e. it is a projection operator. Further, if $\mathcal{G}$ is unitary, then $\langle \omega, \Psi\omega' \rangle = \langle \Psi\omega, \omega' \rangle \; \forall \omega, \omega' \in \mathbb{R}^d$*

**Sample Complexity and Generalization.** On applying the operator $\Psi$ to the dataset $\mathcal{X}$, all points in the set $\{gx \mid g \in \mathcal{G}\}$ for any $x \in \mathcal{X}$ map to the same point $\Psi x$ in the $\mathcal{G}$-invariant subspace thereby reducing the number of distinct points by a factor of $|\mathcal{G}|$ (the cardinality of $\mathcal{G}$, if $\mathcal{G}$ is finite). Theoretically, this would drastically reduce sample complexity while preserving linear feasibility (separability). It is trivial to observe that *a perfect linear separator learned in $\mathcal{X}_\Psi = \{\Psi x \mid x \in \mathcal{X}\}$ would also be a perfect separator for $\mathcal{X}_{\mathcal{G}}$*, thus in theory achieving perfect generalization. Generalization here refers to the ability to perform correct classification even in the presence of the set of transformations $\mathcal{G}$. We prove a similar result for Reproducing Kernel Hilbert Spaces (RKHS) in Section 2.2. This property is theoretically powerful since cardinality of $\mathcal{G}$ can be large. *A classifier can avoid having to observe transformed versions $\{gx\}$ of any $x$ and yet generalize perfectly.*

**The case of Face Recognition.** As an illustration, if the group $\mathcal{G}$ of transformations considered is pose (it is hypothesized that small changes in pose can be modeled as unitary [10]), then $\Psi = \int_{\mathcal{G}} g \, dg$ represents a pose invariant subspace. In theory, all poses of a subject will converge to the same point in that subspace leading to near perfect pose invariant recognition.

We have not yet leveraged the power of the unitary structure of the groups which is also critical in generalization to test cases as we would see later. We now present our central result showcasing that unitary kernels allow the unitary group action to reciprocate in a Reproducing Kernel Hilbert Space. This is critical to set the foundation for our core method called Max-Margin Invariant Features.

## 2.1 Group Actions Reciprocate in a Reproducing Kernel Hilbert Space

Group integration provides exact invariance as seen in the previous section. However, it requires the group structure to be preserved, *i.e.* if the group structure is destroyed, group integration does not provide an invariant function. In the context of kernels, it is imperative that the group relation between the samples in $\mathcal{X}_{\mathcal{G}}$ be preserved in the kernel Hilbert space $\mathcal{H}$ corresponding to some kernel $k$ with a mapping $\phi$. If the kernel $k$ is unitary in the following sense, then this is possible.

**Definition 2.2** (*Unitary Kernel*). A kernel $k(x, y) = \langle \phi(x), \phi(y) \rangle$ is a unitary kernel if, for a unitary group $\mathcal{G}$, the mapping $\phi(x) : \mathcal{X} \to \mathcal{H}$ satisfies $\langle \phi(gx), \phi(gy) \rangle = \langle \phi(x), \phi(y) \rangle \; \forall g \in \mathcal{G}, \forall x, y \in \mathcal{X}$.

The unitary condition is fairly general, a common class of unitary kernels is the RBF kernel. We now define a transformation within the RKHS itself as $g_{\mathcal{H}} : \phi(x) \to \phi(gx) \; \forall \phi(x) \in \mathcal{H}$ for any $g \in \mathcal{G}$ where $\mathcal{G}$ is a unitary group. We then have the following result of significance.

**Theorem 2.4.** *(Covariance in the RKHS) If $k(x, y) = \langle \phi(x), \phi(y) \rangle$ is a unitary kernel in the sense of Definition 2.2, then $g_{\mathcal{H}}$ is a unitary transformation, and the set $\mathcal{G}_{\mathcal{H}} = \{g_{\mathcal{H}} \mid g_{\mathcal{H}} : \phi(x) \to \phi(gx) \; \forall g \in \mathcal{G}\}$ is a unitary-group in $\mathcal{H}$.*

Theorem 2.4 shows that the unitary-group structure is preserved in the RKHS. This paves the way for new theoretically motivated approaches to achieve invariance to transformations in the RKHS. There have been a few studies on group invariant kernels [4, 10]. However, [4] does not examine whether the unitary group structure is actually preserved in the RKHS, which is critical. Also, DIKF was recently proposed as a method utilizing group structure under the unitary kernel [10]. Our result is a generalization of the theorems they present. Theorem 2.4 shows that since the unitary group structure is preserved in the RKHS, any method involving group integration would be invariant in the original space. The preservation of the group structure allows more direct group invariance results to be applied in the RKHS. It also directly allows one to formulate a non-linear SVM while guaranteeing invariance theoretically leading to Max-Margin Invariant Features.

## 2.2 Invariant Non-linear SVM: An Alternate Approach Through Group Integration

We now apply the group integration approach to the kernel SVM. The decision function of SVMs can be written in the general form as $f_\theta(x) = \omega^T \phi(x) + b$ for some bias $b \in \mathbb{R}$ (we agglomerate all parameters of $f$ in $\theta$) where $\phi$ is the kernel feature map, *i.e.* $\phi : \mathcal{X} \to \mathcal{H}$. Reviewing the SVM, a maximum margin separator is found by minimizing loss functions such as the hinge loss along with a regularizer. In order to invoke invariance, we can now utilize group integration in the the kernel space $\mathcal{H}$ using Theorem 2.4. All points in the set $\{gx \in \mathcal{X}_\mathcal{G}\}$ get mapped to $\phi(gx) = g_\mathcal{H}\phi(x)$ for a given $g \in \mathcal{G}$ in the input space $\mathcal{X}$. Group integration then results in a $\mathcal{G}$-invariant subspace within $\mathcal{H}$ through $\Psi_\mathcal{H} = \int_{\mathcal{G}_\mathcal{H}} g_\mathcal{H} \, dg_\mathcal{H}$ using Lemma 2.1. Introducing Lagrange multipliers $\alpha = (\alpha_1, \alpha_2...\alpha_N) \in \mathbb{R}^N$, the dual formulation (utilizing Lemma 2.2 and Lemma 2.3) then becomes

$$\min_\alpha - \sum_i \alpha_i + \frac{1}{2} \sum_{i,j} y_i y_j \alpha_i \alpha_j \langle \Psi_\mathcal{H}\phi(x_i), \Psi_\mathcal{H}\phi(x_j) \rangle \tag{1}$$

under the constraints $\sum_i \alpha_i y_i = 0, \quad 0 \le \alpha_i \le \frac{1}{N} \quad \forall i$. The SVM separator is then given by $\omega_\mathcal{H}^* = \Psi_\mathcal{H}\omega^* = \sum_i y_i \alpha_i \Psi_\mathcal{H}\phi(x_i)$ thereby existing in the $\mathcal{G}_\mathcal{H}$-invariant (or equivalently $\mathcal{G}$-invariant) subspace $\Psi_\mathcal{H}$ within $\mathcal{H}$ (since $g \to g_\mathcal{H}$ is a bijection). Effectively, the SVM observes samples from $\mathcal{X}_{\Psi_\mathcal{H}} = \{x \mid \phi(x) = \Psi_\mathcal{H}\phi(u), \forall u \in \mathcal{X}_\mathcal{G}\}$ and therefore $\omega_\mathcal{H}^*$ enjoys *exact global* invariance to $\mathcal{G}$. Further, $\Psi_\mathcal{H}\omega^*$ *is a maximum-margin separator of* $\{\phi(\mathcal{X}_G)\}$ (*i.e.* the set of all transformed samples). This can be shown by the following result.

**Theorem 2.5.** *(Generalization) For a unitary group $\mathcal{G}$ and unitary kernel $k(x,y) = \langle \phi(x), \phi(y) \rangle$, if $\omega_\mathcal{H}^* = \Psi_\mathcal{H}\omega^* = (\int_{\mathcal{G}_\mathcal{H}} g_\mathcal{H} \, dg_\mathcal{H}) \omega^*$ is a perfect separator for $\{\Psi_\mathcal{H}\phi(\mathcal{X})\} = \{\Psi_\mathcal{H}\phi(x) \mid \forall x \in \mathcal{X}\}$, then $\Psi_\mathcal{H}\omega^*$ is also a perfect separator for $\{\phi(\mathcal{X}_G)\} = \{\phi(x) \mid x \in \mathcal{X}_\mathcal{G}\}$ with the same margin. Further, a max-margin separator of $\{\Psi_\mathcal{H}\phi(\mathcal{X})\}$ is also a max-margin separator of $\{\phi(\mathcal{X}_G)\}$.*

The invariant non-linear SVM in objective 1, observes samples in the form of $\Psi_\mathcal{H}\phi(x)$ and obtains a max-margin separator $\Psi_\mathcal{H}\omega^*$. *This allows for the generalization properties of max-margin classifiers to be combined with those of group invariant classifiers.* While being invariant to nuisance transformations, max-margin classifiers can lead to highly discriminative features (more robust than DIKF [10] as we find in our experiments) that are invariant to within-class transformations.

Theorem 2.5 shows that the margins of $\phi(\mathcal{X}_\mathcal{G})$ and $\{\Psi_\mathcal{H}\phi(\mathcal{X}_\mathcal{G})\}$ are deeply related and implies that $\Psi_\mathcal{H}\phi(x)$ is a max-margin separator for both datasets. Theoretically, the invariant non-linear SVM is able to generalize to $\mathcal{X}_\mathcal{G}$ on *just* observing $\mathcal{X}$ and utilizing prior information in the form of $\mathcal{G}$ for all unitary kernels $k$. This is true *in practice* for linear kernels. For non-linear kernels in practice, the invariant SVM still needs to observe and integrate over transformed **training** inputs.

**Leveraging unitary group properties.** During test time to achieve invariance, the SVM would require to observe and integrate over all possible transformations of the test sample. This is a huge computational and design bottleneck. We would ideally want to achieve invariance and generalize by observing just a single test sample, in effect perform *one shot learning*. This would not only be computationally much cheaper but make the classifier powerful owing to generalization to full transformed orbits of test samples by observing just that single sample. This is where unitarity of $g$ helps and we leverage it in the form of the following Lemma.

**Lemma 2.6.** *(Invariant Projection) If $\Psi = \int_\mathcal{G} g \, dg$ for any unitary group $\mathcal{G}$, then for any fixed $g' \in \mathcal{G}$ (including the identity element) we have $\langle \Psi x', \Psi \omega' \rangle = \langle g'x', \Psi \omega' \rangle \; \forall \omega, \omega' \in \mathbb{R}^d$*

Assuming $\Psi \omega'$ is the learned SVM classifier, Lemma 2.6 shows that for any *test* $x'$, the invariant dot product $\langle \Psi x', \Psi \omega' \rangle$ which involves observing *all transformations* of $x'$ is equivalent to the quantity $\langle g'x', \Psi \omega' \rangle$ which involves observing *only one* transformation of $x'$. Hence *one can model the entire orbit of $x'$ under $\mathcal{G}$ by a single sample $g'x'$ where $g' \in \mathcal{G}$ can be any particular transformation including identity.* This drastically reduces sample complexity and vastly increases generalization capabilities of the classifier since one only need to observe one test sample to achieve invariance Lemma 2.6 also helps us in saving computation, allowing us to apply the computationally expensive $\Psi$ (group integration) operation only once on he classifier and not the test sample. Thus, the kernel in the Invariant SVM formulation can be replaced by the form $k_\Psi(x,y) = \langle \phi(x), \Psi_\mathcal{H}\phi(y) \rangle$.

For kernels in general, the $\mathcal{G}_\mathcal{H}$-invariant subspace cannot be explicitly computed since it lies in the RKHS. It is only implicitly projected upon through $\Psi_\mathcal{H}\phi(x_i) = \int_\mathcal{G} \phi(gx_i)dg_\mathcal{H}$. It is important to

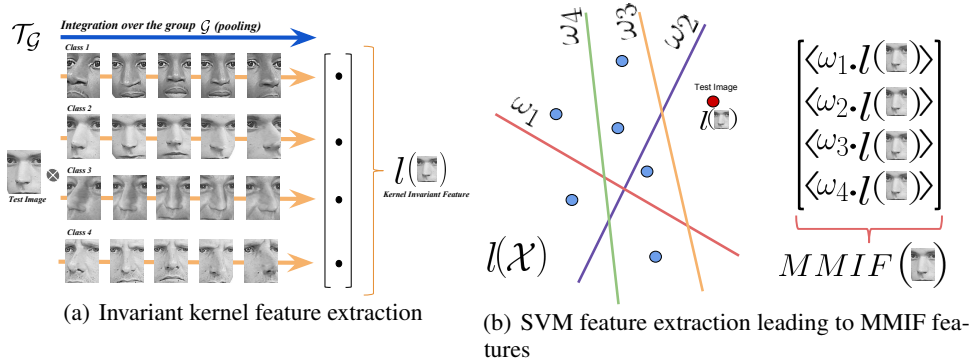

(a) Invariant kernel feature extraction

(b) SVM feature extraction leading to MMIF features

**Figure 2: MMIF Feature Extraction.** (a) $l(x)$ denotes the invariant kernel feature of any $x$ which is invariant to the transformation $\mathcal{G}$. Invariance is generated by group integration (or pooling). The invariant kernel feature learns invariance form the *unlabeled* transformed template set $\mathcal{T}_\mathcal{G}$. Also, the faces depicted are actual samples from the large-scale mugshots data ($\sim 153,000$ images). (b) Once the invariant features have been extracted for the *labelled non-transformed* dataset $\mathcal{X}$, then the SVMs learned act as feature extractors. Each binary class SVM (different color) was trained on the invariant kernel feature of a random subset of $l(\mathcal{X})$ with random class assignments. The final MMIF feature for $x$ is the concatenation of all SVM inner-products with $l(x)$.

note that during *testing* however, the SVM formulation will be invariant to transformations of the test sample regardless of a linear or non-linear kernel.

**Positive Semi-Definiteness.** The $\mathcal{G}$-invariant kernel map is now of the form $k_\Psi(x,y) = \langle \phi(x), \int_\mathcal{G} \phi(gy)dg_\mathcal{H} \rangle$. *This preserves the positive semi-definite property of the kernel $k$ while guaranteeing global invariance to unitary transformations.*, unlike jittering kernels [17, 3] and tangent-distance kernels [5]. If we wish to include invariance to *scaling* however (in the sense of scaling an image), then we would lose positive-semi-definiteness (it is also not a unitary transform). Nonetheless, [20] show that conditionally positive definite kernels still exist for transformations including scaling, although we focus of unitary transformations in this paper.

## 3 Max-Margin Invariant Features

The previous section utilized a group integration approach to arrive a theoretically invariant non-linear SVM. It however does not address the Unlabeled Transformation problem *i.e.* the kernel $k_\Psi(x,y) = \langle \Psi_\mathcal{H}\phi(x), \Psi_\mathcal{H}\phi(y) \rangle = \langle \int_\mathcal{G} \phi(gx)dg_\mathcal{H}, \int_\mathcal{G} \phi(gy)dg_\mathcal{H} \rangle$ still requires observing transformed versions of the *labelled* input sample namely $\{gx \mid gx \in \mathcal{X}_\mathcal{G}\}$ (or atleast one of the labelled samples if we utilize Lemma 2.6). We now present our core approach called Max-Margin Invariant Features (MMIF) that does not require the observation of any *transformed labelled* training sample whatsoever.

Assume that we have access to an *unlabeled* set of $M$ templates $\mathcal{T} = \{t_i\}_{i=\{1,...M\}}$. We assume that we can observe all transformations under a unitary-group $\mathcal{G}$, *i.e.* we have access to $\mathcal{T}_\mathcal{G} = \{gt_i \mid \forall g \in \mathcal{G}\}_{i=\{1,...M\}}$. Also, assume we have access to a set $\mathcal{X} = \{x_j\}_{i=\{1,...D\}}$ of *labelled* data with $N$ classes which are **not** transformed. We can extract an $M$-dimensional invariant kernel feature for each $x_j \in \mathcal{X}$ as follows. Let the invariant kernel *feature* be $l(x) \in \mathbb{R}^M$ to explicitly show the dependence on $x$. Then the $i^{th}$ dimension of $l$ for any particular $x$ is computed as

$$l(x)_i = \langle \phi(x), \Psi_\mathcal{H}\phi(t_i) \rangle = \langle \phi(x), \int_\mathcal{G} g_\mathcal{H}\phi(t_i)dg_\mathcal{H} \rangle = \langle \phi(x), \int_\mathcal{G} \phi(gt_i)dg_\mathcal{H} \rangle \qquad (2)$$

The first equality utilizes Lemma 2.6 and the third equality uses Theorem 2.4. This is equivalent to observing all transformations of $x$ since $\langle \phi(x), \Psi_\mathcal{H}\phi(t_i) \rangle = \langle \Psi_\mathcal{H}\phi(x), \phi(t_i) \rangle$ using Lemma 2.3. Thereby we have constructed a feature $l(x)$ which is invariant to $\mathcal{G}$ without ever needing to observe transformed versions of the labelled vector $x$. We now briefly the training of the MMIF feature extractor. The matching metrics we use for this study is normalized cosine distance.

**Training MMIF SVMs.** To learn a $K$-dimensional MMIF feature (potentially independent of $N$), we learn $K$ independent binary-class linear SVMs. Each SVM trains on the labelled dataset $l(\mathcal{X}) = \{l(x_j) \mid j = \{1, ...D\}\}$ with each sample being label $+1$ for some subset of the $N$ classes (potentially just one class) and the rest being labelled $-1$. This leads us to a classifier in the form of $\omega_k = \sum_j y_j \alpha_j l(x_j)$. Here, $y_j$ is the label of $x_j$ for the $k^{th}$ SVM. *It is important to note that the unlabeled data was only used to extract* $l(x_j)$. Having multiple classes randomly labelled as positive allows the SVM to extract some feature that is common between them. This increases generalization by forcing the extracted feature to be more general (shared between multiple classes) rather than being highly tuned to a single class. Any $K$-dimensional MMIF feature can be trained through this technique leading to a higher dimensional feature vector useful in case where one has limited labelled samples and classes ($N$ is small). During feature extraction, the $K$ inner products (scores) of the test sample $x'$ with the $K$ distinct binary-class SVMs provides the $K$-dimensional MMIF feature vector. This feature vector is highly discriminative due to the max-margin nature of SVMs while being invariant to $\mathcal{G}$ due to the invariant kernels.

**MMIF.** Given $\mathcal{T}_\mathcal{G}$ and $\mathcal{X}$, the MMIF feature is defined as $\text{MMIF}(x') \in \mathbb{R}^K$ for any test $x'$ with each dimension $k$ being computed as $\langle l(x'), \omega_k \rangle$ for $\omega_k = \sum_j y_j \alpha_j l(x_j) \; \forall x_j \in \mathcal{X}$. Further, $l(x') \in \mathbb{R}^M \; \forall x$ with each dimension $i$ being $l(x')_i = \langle \phi(x'), \Psi_\mathcal{H} \phi(t_i) \rangle$. The process is illustrated in Fig. 2.

**Inheriting transformation invariance from transformed unlabeled data: A special case of semi-supervised learning.** MMIF features can learn to be invariant to transformations ($\mathcal{G}$) by observing them *only* through $\mathcal{T}_\mathcal{G}$. It can then transfer the invariance knowledge to new unseen samples from $\mathcal{X}$ thereby becoming invariant to $\mathcal{X}_\mathcal{G}$ despite never having observed any samples from $\mathcal{X}_\mathcal{G}$. This is a special case of semi-supervised learning where we leverage on the specific transformations present in the unlabeled data. This is a very useful property of MMIFs allowing one to learn transformation invariance from one source and sample points from another source while having powerful discrimination and generalization properties. The property is can be formally stated as the following Theorem.

**Theorem 3.1.** *(MMIF is invariant to learnt transformations)* $\text{MMIF}(x') = \text{MMIF}(gx') \; \forall x' \forall g \in \mathcal{G}$ *where* $\mathcal{G}$ *is observed only through* $\mathcal{T}_\mathcal{G} = \{gt_i \mid \forall g \in \mathcal{G}\}_{i=\{1,...M\}}$.

*Thus we find that MMIF can solve the Unlabeled Transformation Problem.* MMIFs have an invariant and a discriminative component. The invariant component of MMIF allows it to generalize to new transformations of the test sample whereas the discriminative component allows for robust classification due to max-margin classifiers. These two properties allow MMIFs to be very useful as we find in our experiments on face recognition.

**Max and Mean Pooling in MMIF.** Group integration in practice directly results in mean pooling. Recent work however, showed that group integration can be treated as a subset of I-theory where one tries to measure moments (or a subset of) of the distribution $\langle x, g\omega \rangle \; g \in \mathcal{G}$ since the distribution itself is also an invariant [1]. Group integration can be seen as measuring the mean or the first moment of the distribution. One can also characterize using the infinite moment or the max of the distribution. We find in our experiments that max pooling outperforms mean pooling in general. All results in this paper however, still hold under the I-theory framework.

**MMIF on external feature extractors (deep networks).** MMIF does not make any assumptions regarding its input and hence one can apply it to features extracted from any feature extractor in general. The goal of any feature extractor is to (ideally) be invariant to within-class transformation while maximizing between-class discrimination. However, most feature extractors are not trained to explicitly factor out specific transformations. If we have access to even a small dataset with the transformation we would like to be invariant to, we can *transfer* the invariance using MMIFs (*e.g.* it is unlikely to observe all poses of a person in datasets, but pose is an important nuisance transformation).

**Modelling general non-unitary transformations.** General non-linear transformations such as out-of-plane rotation or pose variation are challenging to model. Nonetheless, a small variation in these transformations can be approximated by some unitary $\mathcal{G}$ assuming piece wise linearity through transformation-dependent sub-manifold unfolding [11]. Further, it was found that in practice, integrating over general transformations produced approximate invariance [8].

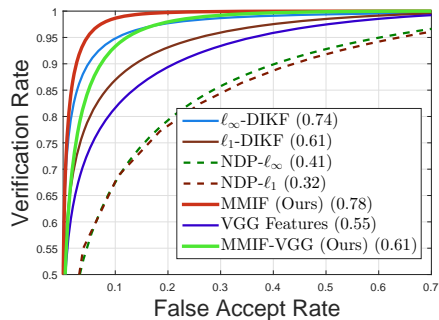 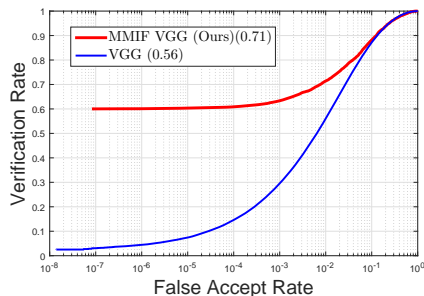

(a) Invariant kernel feature extraction

(b) SVM feature extraction leading to MMIF features

**Figure 3:** (a) Pose-invariant face recognition results on the semi-synthetic large-scale mugshot database (testing on 114,750 images). **Operating on pixels:** MMIF (Pixels) outperforms invariance based methods DIKF [10] and invariant NDP [8]. **Operating on deep features:** MMIF trained on VGG-Face features [12] (MMIF-VGG) produces a significant improvement in performance. The numbers in the brackets represent VR at $0.1\%$ FAR. (b) Face recognition results on LFW with raw VGG-Face features and MMIF trained on VGG-Face features. The values in the bracket show VR at $0.1\%$ FAR.

## 4 Experiments on Face Recognition

As illustration, we apply MMIFs using two modalities overall 1) on raw pixels and 2) on deep features from the pre-trained VGG-Face network [12]. We provide more implementation details and results discussion in the supplementary.

**A. MMIF on a large-scale semi-synthetic mugshot database (Raw-pixels and deep features).** We utilize a large-scale semi-synthetic face dataset to generate the sets $\mathcal{T}_{\mathcal{G}}$ and $\mathcal{X}$ for MMIF. In this dataset, only two major transformations exist, that of pose variation and subject variation. All other transformations such as illumination, translation, rotation etc are strictly and synthetically controlled. This provides a very good benchmark for face recognition. where we want to be invariant to pose variation and be discriminative for subject variation. The experiment follows the exact protocol and data as described in [10] [3] We test on 750 subjects identities with 153 pose varied real-textured gray-scale image each (a total of 114,750 images) against each other resulting in about 13 billion pair-wise comparisons (compared to 6,000 for the standard LFW protocol). Results are reported as ROC curves along with VR at $0.1\%$ FAR. Fig. 3(a) shows the ROC curves for this experiment. We find that MMIF features out-performs all baselines including VGG-Face features (pre-trained), DIKF and NDP approaches thereby demonstrating superior discriminability while being able to effectively capture pose-invariance from the transformed template set $\mathcal{T}_{\mathcal{G}}$. MMIF is able to solve the Unlabeled Transformation problem by extracting transformation information from unlabeled $\mathcal{T}_{\mathcal{G}}$.

**B. MMIF on LFW (deep features): Unseen subject protocol.** In order to be able to effectively train under the scenario of general transformations and to challenge our algorithms, we define a new much harder protocol on LFW. We choose the top 500 subjects with a total of 6,300 images for training MMIF on VGG-Face features and test on the remaining subjects with 7,000 images. We perform all versus all matching, totalling upto 49 million matches (4 orders more than the official protocol). The evaluation metric is defined to be the standard ROC curve with verification rate reported at $0.1\%$ false accept rate. We split the 500 subjects into two sets of 250 and use as $\mathcal{T}_{\mathcal{G}}$ and $\mathcal{X}$. We do not use any alignment for this experiment, and the faces were cropped according to [16]. Fig. 3(b) shows the results of this experiment. We see that MMIF on VGG features significantly outperforms raw VGG on this protocol, boosting the VR at $0.1\%$ FAR from 0.56 to 0.71. This demonstrates that MMIF is able to generate invariance for highly non-linear transformations that are not well-defined rendering it useful in real-world scenarios where transformations are unknown but observable.

## Footnotes

[2] All proofs are presented in the supplementary material

[3]We provide more details in the supplementary. Also note that we do not need utilize identity information, all that is required is the fact that a set of pose varied images belong to the same subject. Such data can be obtained through temporal sampling.

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
