[Supplementary Material]

# Max-Margin Invariant Features from Transformed Unlabeled Data: Supplementary Material

**Dipan K. Pal, Ashwin A. Kannan,** *** Gautam Arakalgud,** *** Marios Savvides**
Department of Electrical and Computer Engineering
Carnegie Mellon University
Pittsburgh, PA 15213
{dipanp,aalapakk,garakalgud,marioss}@cmu.edu

## Abstract

We provide more details regarding the main experiments presented in the main paper. We also present additional results on the large scale semi-synthetic data and finally present results of a short ablation study on a separate protocol for the IARPA IJB-A Janus dataset. Note that throughout this supplementary and the main paper we always test on completely unseen subjects. On the theoretical side, we provide proofs of the analytical results in the main paper.

## 1 Main Experiments: Detailed notes supplementing the main paper.

**A. MMIF on a large-scale semi-synthetic mugshot database (Raw-pixels and deep features).**

**MMIF template set $\mathcal{T}_{\mathcal{G}}$ and $\mathcal{X}$.** We utilize a large-scale semi-synthetic face dataset to generate the sets $\mathcal{T}_{\mathcal{G}}$ and $\mathcal{X}$ for MMIF. The face textures are sampled from real-faces although the poses are rendered using 3D model fit to each face independently, hence the dataset is semi-synthetic. This semi-synthetic dataset helps us to evaluate our algorithm in a clean setting, where there exists only one challenging nuisance transformation (*pose variation*). Therefore $\mathcal{G}$ models pose variation in faces. We utilize the same pose variation dataset generation procedure as described in [4] in order for a fair comparison. The poses were rendered varying from $-40°$ to $40°$ (yaw) and $-20°$ to $20°$ (pitch) in steps of $5°$ using 3D-GEM [6]. The total number of images we generate is $153 \times 1000 = 153,000$ images. We align all faces by the two eye-center locations in a $168 \times 128$ crop.

**Protocol.** Our first experiment is a direct comparison with approaches similar in spirit to ours, namely $\ell_{\infty}$-DIKF and $\ell_1$-DIKF [4] and NDP-$\ell_{\infty}$ and NDP-$\ell_1$ [3, 1]. We train on 250 subjects (38,250 images) and test each method on the remaining 750 subjects (114,750 images), matching all pose-varied images of a subject to each other. DIKF follows the same protocol as in [4]. For MMIF, we utilize the first $125 \times 153$ images (125 subjects with 153 poses each) as $\mathcal{T}_{\mathcal{G}}$ and the next $125 \times 153$ images as $\mathcal{X}$. A total of 500 SVMs were trained on subsets of $\mathcal{X}$ (10 randomly chosen subjects per SVM with all images of 3 of those 10 subjects, again randomly chosen, being $+1$ and the rest being $-1$). Note that although $\mathcal{X}$ in this case contains pose variation, we do not integrate over them to generate invariance. All explicit invariance properties are generated through integration over $\mathcal{T}_{\mathcal{G}}$. For testing, we compare all 153 images of the remaining *unseen* 750 subjects against each other (114,750 images). The algorithms are therefore tested on about 13 billion pair wise comparisons. Results are reported as ROC curves along with VR at $0.1\%$ FAR. For this experiment, we report results working on 1) raw pixels directly and 2) 4096 dimensional features from the pre-trained VGG-Face network [5]. As a baseline, we also report results on using the VGG-Face features directly.

**Results.** Fig.3(a) shows the ROC curves for this experiment. We find that MMIF features out-perform both DIKF and NDP approaches thereby demonstrating superior discriminability while being able to

---

effectively capture pose-invariance from the transformed template set $\mathcal{T_G}$. We find that VGG-Face features suffer a handicap due to the images being grayscale. Nonetheless, MMIF is able to transfer pose-invariance from $\mathcal{T_G}$ onto the VGG features. This significantly boosts performance owing to the fact that the main nuisance transformation is pose. MMIF being explicitly pose invariant along with solving the Unlabeled Transformation Problem is able to help VGG features while preserving the discriminability of the VGG features. In fact, the max-margin SVMs further add discriminability. This illustrates in a clean setting (dataset only contains synthetically generated pose variation as nuisance transformation), that MMIF is able to work well in conjunction with deep learning features, thereby rendering itself immediately usable in more realistic settings. Our next set of experiments focus on this exact aspect.

**B. MMIF on LFW (deep features).**

**Unseen subject protocol.** LFW [2] has received a lot of attention in the recent years, and algorithms have approached near human accuracy on the original testing protocol. In order to be able to effectively train under the scenario of general transformations and to challenge our algorithms, we define a new much harder protocol on LFW. Instead of evaluating on about 6000 pair wise matches, we pair wise match on all images of subjects *not seen in training*. We have no way of modelling these subjects whatsoever, making this a difficult task. We utilize 500 subjects and all their images for training and test on the remaining 5249 subjects and all of their images. To use maximum amount of data for training, we pick the top 500 subjects with the most number of images available (about 6,300 images). The test data thus contains about 7000 images. The number of test pairwise matches is about 49 million, four orders of magnitude larger than the 6000 matches that the original LFW testing protocol defined. The evaluation metric is defined to be the standard ROC curve with verification rate reported at $0.1\%$ false accept rate.

**MMIF template set $\mathcal{T_G}$ and $\mathcal{X}$.** We split the 500 subjects data into two parts of 250 subjects each. We use the 250 subjects with the most number of images as transformed template set $\mathcal{T_G}$ and use the rest of the 250 subjects as $\mathcal{X}$. Note that in this experiment, the transformations considered are very generic and highly non-linear making it a difficult experiment. We do not use any alignment for this experiment, and the faces were cropped according to [7].

**Protocol.** For MMIF, we process the kernel features from the transformed template set $\mathcal{TG}$ exactly as in the previous experiment A. Similarly, we learn a total of 500 SVMs on subsets of $\mathcal{X}$ following the same protocol as the previous experiment.

**Results.** Fig.3(b) shows the results of this experiment. We see that MMIF on VGG features significantly outperforms raw VGG on this protocol, boosting the VR at $0.1\%$ FAR from 0.56 to 0.71. This suggests, that MMIF can be used in conjunction with pre-trained deep features. In this experiment, MMIF capitalizes on the non-linear transformations that exist in LFW, whereas in the previous experiment on the semi0synthetic dataset (Experiment A), the transformation was well-defined to be pose variation. This demonstrates that MMIF is able to generate invariance for highly non-linear transformations that are not well-defined rendering it useful in real-world scenarios where transformations are unknown but observable.

## 2  Additional Experiments

### 2.1  Large-scale Semi Synthetic Mugshot Data

**Motivation:** In the main paper, the transformations were observed only through unlabeled $\mathcal{T_G}$ while $\mathcal{X}$ is only meant to provide labeled untransformed data. However, during our experiments in the main paper, even though we do not explicitly pool over the transformations $\mathcal{X}$, we utilize all transformations for training the SVMs. In order to be closer to our theoretical setting, we now run MMIF on raw pixels and VGG-Face features [5] while *constraining* the number of images the SVMs train on to 30 random images for each subject.

**MMIF Template set $\mathcal{T_G}$ and $\mathcal{X}$:** We utilize a large scale semi-synthetic face dataset to generate the template set $\mathcal{T_G}$ for MMIF. The face textures are sampled from real faces and the poses are rendered using a 3D model fit to each face independently, making the dataset semi-synthetic. This semi-synthetic dataset helps us evaluate our algorithm in a clean setting, where there exists only one challenging nuisance transformation (pose variation). Therefore $\mathcal{G}$ models pose variation in faces. We utilize the same pose variation dataset generation procedure as described in [4] in order for a fair

comparison. The poses were rendered varying from $-40\circ$ to $40\circ$ (yaw) and $-20\circ$ to $20\circ$ (pitch) in steps of $5\circ$ using 3D-GEM [15]. The total number of images we generate is 153 x 1000 = 153,000 images. We align all faces by the two eye-center locations in a $168 \times 128$ crop. *Unlike our experiment presented in the main paper on this dataset, the template set $\mathcal{X}$ is constrained to include only 30 randomly selected poses that $\mathcal{T}_\mathcal{G}$ contained* . This is done to better simulate a real-world setting where through $\mathcal{X}$ we would only observe faces at a few random poses.

**Protocol:** This experiment is a direct comparison with approaches similar in spirit to ours, namely $l_\infty$-DIKF and $l_1$-DIKF [4] and NDP-$l_\infty$ and NDP-$l_1$ [3, 1]. We call this setting for MMIF as *MMIF-cons (constrained)* for reference. We train on 250 subjects (38,250 images) and test each method on the remaining 750 subjects (114,750 images), matching all pose-varied images of a subject to each other. DIKF follows the same protocol as in [4].

For MMIF, we utilize the first 125 x 153 images (125 subjects with 153 poses each) as the template set $\mathcal{T}_\mathcal{G}$. Thus, $\mathcal{T}_\mathcal{G}$ remains exactly the same as the protocol in the main paper. The template set $\mathcal{X}$ is generated by choosing 30 random poses (for every subject) of the next 125 subjects. A total of 500 SVMs are trained on $\mathcal{X}$ with a random subset of 5 subjects being labeled +1 and the rest labeled -1. It's important to note that since $\mathcal{X}$ does not contain transformations that are observed in its entirety, all explicit invariance properties are generated through integration over $\mathcal{T}_\mathcal{G}$.

For testing, we follow the same protocol as in the main paper. We compare all 153 images of the remaining unseen 750 subjects against each other (114,750 images). The algorithms

**Figure 1:** Pose-invariant face recognition results on the semi-synthetic large-scale mugshot database (testing on 114,750 images). **Operating on deep features:** MMIF-cons-VGG trained on VGG-Face features [5] produces a significant improvement in performance over pure VGG features even though it utilizes a constrained $\mathcal{X}$ set. Interestingly, MMIF-cons-VGG almost matches performance of MMIF-VGG while using less data. The numbers in the brackets represent VR at $0.1\%$ FAR. MMIF-cons was trained on the entire $\mathcal{T}_\mathcal{G}$ but only 30 random transformations per subject in the $\mathcal{X}$.

are therefore tested on about 13 billion pair wise comparisons. Results are reported as ROC curves along with the VR at 0.1% FAR. For this experiment, we report results working on 1) raw pixels directly and 2) 4096 dimensional features from the pre-trained VGG-Face network [5]. As a baseline, we also report results on using the VGG-Face features directly.

**Results:** Fig. 1 shows the ROC curves for this experiment. We find that even though we train SVMs for MMIF-cons-VGG on a constrained version of $\mathcal{X}$, it outperforms raw VGG features. Although, we do observe that MMIF-cons-raw outperforms NDP methods thereby demonstrating superior discriminability, it fails to match the original MMIF-raw method performance. Interestingly however, MMIF-cons-VGG matches MMIF-VGG features in performance despite being trained on much lesser data (30 instead of 153 images per subject). Thus, we find that MMIF when trained on a good feature extractor can provide added benefits of discrimination despite having lesser labeled samples to train on.

## 2.2 IARPA IJB-A Janus

In this experiment, we explore how the number of SVMs influences the recognition performance on a large scale real-world dataset, namely the IARPA Janus Benchmark A (IJB-A) dataset.

**Data:** We work on the verification protocol (1:1 matching) of the original dataset IJB-A Janus. This subset consists of 5547 image templates that map to 492 distinct subjects with each template containing (possibly) multiple images. The images are cropped with respect to bounding boxes that are specified by the dataset for all labeled images. The cropped images are then re-sized to 244 x 244 pixels in accordance with the requirements of the VGG face model. Explicit pose invariance (MMIF) is then applied to these general face descriptors.

**MMIF Template set $\mathcal{T}_{\mathcal{G}}$ and $\mathcal{X}$:** In order to effectively train under the scenario of general transformations, we define a new protocol the Janus dataset similar to the LFW protocol defined in the main paper. This protocol is suited for MMIF since we explicitly generate invariance to transformations that exist in Janus data. We utilize the first 100 subjects and all the templates that map to these subjects (23723 images) for training MMIF and test on the remaining 392 subjects (27363 images). To make use of the maximum amount of data for training, we pick the top 100 subjects with the most number of images, the rest are all utilized for testing. Our training dataset is further split into templates $\mathcal{T}_{\mathcal{G}}$ and $\mathcal{X}$ similar to our LFW protocol in the main paper. We use the first 50 subjects (of the top 100 subjects) as $\mathcal{T}_{\mathcal{G}}$ and the rest as $\mathcal{X}$ in order to maximize the transformations that we generate invariance towards. To showcase the ability of MMIF to be used in conjunction with deep learning techniques, similar to our LFW experiment in the main paper, we train and test on VGG-Face features [5] on the Janus data.

**Protocol:** As in our LFW experiment, we split the training data into two templates - $\mathcal{T}_{\mathcal{G}}$ and $\mathcal{X}$. Similarly to all MMIF protocols in this paper, we train a total of 100, 250 and 500 SVM's on subsets of $\mathcal{X}$ following the same protocol. We perform pairwise comparisons for the entirety of the test data ($\sim 750$ million image comparisons) which far exceeds the number of comparisons defined in the original testing protocol ($\sim 110,000$ template comparisons) thereby making this protocol much larger and harder. Recall that throughout this supplementary and the main paper we always test on completely unseen subjects. The evaluation metric is defined to be the standard ROC curve using cosine distance.

**Figure 2:** Results of MMIF trained on VGG-Face features on the IARPA IJB-A Janus dataset for 100, 250 and 500 SVMs. The number in the bracket denotes VR at 0.1% FAR.

**Results:** Fig. 2 shows the ROC curves for this experiment with new much larger and harder protocol. We find that even with just 100 SVMs or 100 max-margin feature extractors, the performance is close to that of 500 feature extractors. This suggests, that though the SVMs provide enough discrimination, the invariant kernel provides bulk of the recognition performance by explicitly being invariant to the transformations in the $\mathcal{T}_{\mathcal{G}}$. Hence, our proposed invariant kernel is effective at learning invariance towards transformations present in a unlabeled dataset. We provide these curves as baselines for future work focusing on the problem on learning unlabeled transformations from a given dataset.

# 3 Proofs of theoretical results

## 3.1 Proof of Lemma 2.1

*Proof.* We have,

$$g' \int_{\mathcal{G}} g\omega \, dg = \int_{\mathcal{G}} g'g\omega \, dg = \int_{\mathcal{G}} g''\omega \, dg'' = \int_{\mathcal{G}} g\omega \, dg$$

Since the normalized Haar measure is invariant, *i.e.* $dg = dg'$. Intuitively, $g'$ simply rearranges the group integral owing to elementary group properties. □

## 3.2 Proof of Lemma 2.2

*Proof.* We have,

$$\Psi^T = (\int_{\mathcal{G}} g \, dg)^T = \int_{\mathcal{G}} g^T \, dg = \int_{\mathcal{G}} g^{-1} \, dg^{-1} = \Psi$$

Using the fact $g \in \mathcal{G} \Rightarrow g^{-1} \in \mathcal{G}$ and $dg = dg^{-1}$. □

### 3.3 Proof Lemma 2.3

*Proof.* We have,

$$\Psi\Psi = \int_{\mathcal{G}}\int_{\mathcal{G}} gh\, dg\, dh \tag{1}$$

$$= \int_{\mathcal{G}}\int_{\mathcal{G}} g'dg'dh \tag{2}$$

$$= \int_{\mathcal{G}} dh \int_{\mathcal{G}} g'dg' \tag{3}$$

$$= \Psi \tag{4}$$

Since the Haar measure is normalized ($\int_{\mathcal{G}} dg = 1$), and invariant. Also for any $\omega, \omega' \in \mathbb{R}^d$, we have
$\langle \omega, \Psi\omega' \rangle = \int_{\mathcal{G}}\langle \omega, g\omega' \rangle dg = \int_{\mathcal{G}}\langle g^{-1}\omega, \omega' \rangle dg^{-1} = \langle \Psi\omega, \omega' \rangle$ $\qquad\square$

### 3.4 Proof of Theorem 2.4

*Proof.* We have $\langle \phi(gx), \phi(gy) \rangle = \langle \phi(x), \phi(y) \rangle = \langle g_{\mathcal{H}}\phi(x), g_{\mathcal{H}}\phi(y) \rangle$, since the kernel $k$ is unitary. Here we define $g_{\mathcal{H}}\phi(x)$ as the action of $g_{\mathcal{H}}$ on $\phi(x)$. Thus, the mapping $g_{\mathcal{H}}$ preserves the dot-product in $\mathcal{H}$ while reciprocating the action of $g$. This is one of the requirements of a unitary operator, however $g_{\mathcal{H}}$ needs to be linear. We note that linearity of $g_{\mathcal{H}}$ can be derived from the linearity of the inner product and its preservation under $g_{\mathcal{H}}$ in $\mathcal{H}$. Specifically for an arbitrary vector $p$ and a scalar $\alpha$, we have

$$||\alpha g_{\mathcal{H}}p - g_{\mathcal{H}}(\alpha p)||^2 \tag{5}$$

$$= \langle \alpha g_{\mathcal{H}}p - g_{\mathcal{H}}(\alpha p), \alpha g_{\mathcal{H}}p - g_{\mathcal{H}}(\alpha p) \rangle \tag{6}$$

$$= ||\alpha g_{\mathcal{H}}p||^2 + ||g_{\mathcal{H}}(\alpha p)||^2 - 2\langle \alpha g_{\mathcal{H}}p, g_{\mathcal{H}}(\alpha p) \rangle \tag{7}$$

$$= |\alpha|||p||^2 + ||\alpha p||^2 - 2\alpha^2\langle p, p \rangle = 0 \tag{8}$$

Similarly for vectors $p, q$, we have $||g_{\mathcal{H}}(p + q) - (g_{\mathcal{H}}p + g_{\mathcal{H}}q)||^2 = 0$

We now prove that the set $\mathcal{G}_{\mathcal{H}}$ is a group. We start with proving the closure property. We have for any fixed $g_{\mathcal{H}}, g'_{\mathcal{H}} \in \mathcal{G}_{\mathcal{H}}$

$$g_{\mathcal{H}}g'_{\mathcal{H}}\phi(x) = g_{\mathcal{H}}\phi(g'x) = \phi(gg'x) = \phi(g''x) = g''_{\mathcal{H}}\phi(x)$$

Since $g'' \in \mathcal{G}$ therefore $g''_{\mathcal{H}} \in \mathcal{G}_{\mathcal{H}}$ by definition. Also, $g_{\mathcal{H}}g'_{\mathcal{H}} = g''_{\mathcal{H}}$ and thus closure is established. Associativity, identity and inverse properties can be proved similarly. The set $\mathcal{G}_{\mathcal{H}} = \{g_{\mathcal{H}} \mid g_{\mathcal{H}} : \phi(x) \rightarrow \phi(gx)\ \forall g \in \mathcal{G}\}$ is therefore a unitary-group in $\mathcal{H}$. $\qquad\square$

### 3.5 Proof of Theorem 2.5

*Proof.* Since $\Psi_{\mathcal{H}}\omega^*$ is a perfect separator for $\{\Psi_{\mathcal{H}}\phi(\mathcal{X})\}$, $\exists\rho' > 0$, s.t. $\min_i y_i(\Psi_{\mathcal{H}}\phi(x_i))^T(\Psi_{\mathcal{H}}\omega^*) \geq \rho'\ \forall\{x_i, y_i\} \in \mathcal{X}$.

Using Lemma 2.4 and Theorem 2.5, we have for any fixed $g'_{\mathcal{H}} \in \mathcal{G}_{\mathcal{H}}$,

$$(\Psi_{\mathcal{H}}\phi(x_i))^T(\Psi_{\mathcal{H}}\omega^*) = (g'_{\mathcal{H}}\phi(x_i))^T(\Psi_{\mathcal{H}}\omega^*)$$

Hence,

$$\min_i y_i(g'_{\mathcal{H}}\phi(x_i))^T(\Psi_{\mathcal{H}}\omega^*) \tag{9}$$

$$= \min_i y_i(\Psi_{\mathcal{H}}\phi(x_i))^T(\Psi_{\mathcal{H}}\omega^*) \geq \rho'\ \forall(g'_{\mathcal{H}} \Rightarrow g) \in \mathcal{G} \tag{10}$$

Thus, $\Psi_{\mathcal{H}}\omega^*$ is perfect separator for $\{\phi(\mathcal{X}_{\mathcal{G}})\}$ with a margin of at-least $\rho'$. It also implies that a max-margin separator of $\{\Psi_{\mathcal{H}}\phi(\mathcal{X})\}$ is also a max-margin separator of $\{\phi(\mathcal{X}_G)\}$. $\qquad\square$

## 3.6 Proof of Lemma 2.6

*Proof.* We have $\langle \Psi x', \Psi \omega' \rangle = \langle \int_g gx', \Psi \omega' \rangle dg = \langle \int_g g'x', \Psi \omega' \rangle dg = \langle g'x', \Psi \omega' \rangle \int_g dg = \langle g'x', \Psi \omega' \rangle$

In the second equality, we fix *any* group element $g' \in \mathcal{G}$ since the inner-product is invariant using the argument $\langle \omega, \Psi \omega' \rangle = \langle g'\omega, \Psi \omega' \rangle$. This is true using Lemma 2.1 and the fact that $\mathcal{G}$ is unitary. Further, the final equality utilizes the fact that the Haar measure $dg$ is normalized. □

## 3.7 Proof of Theorem 3.1

*Proof.* Given $\mathcal{T}_{\mathcal{G}}$ and $\mathcal{X}$, the MMIF feature is defined as $\mathrm{MMIF}(x') \in \mathbb{R}^K$ for any test $x'$ with each dimension $k$ being computed as $\langle l(x'), \omega_k \rangle$ for $\omega_k = \sum_j y_j \alpha_j l(x_j) \ \forall x_j \in \mathcal{X}$. Further, $l(x') \in \mathbb{R}^M \ \forall x$ with each dimension $i$ being $l(x')_i = \langle \phi(x'), \Psi_{\mathcal{H}} \phi(t_i) \rangle$. Here, $\Psi_{\mathcal{H}} = \int_{\mathcal{G}_{\mathcal{H}}} g_{\mathcal{H}} dg_{\mathcal{H}}$ where $g_{\mathcal{H}}$ in the RKHS corresponds to the group action of $g \in \mathcal{G}$ acting in the space of $\mathcal{X}$.

We therefore have for the $i^{th}$ dimension of $l(x')$,

$$l(x')_i = \langle \phi(x'), \Psi_{\mathcal{H}} \phi(t_i) \rangle \tag{11}$$

$$= \langle \phi(x'), \int_{\mathcal{G}_{\mathcal{H}}} g_{\mathcal{H}} \phi(t_i) dg_{\mathcal{H}} \rangle \tag{12}$$

$$= \langle \phi(x'), \int_{\mathcal{G}_{\mathcal{H}}} g_{\mathcal{H}}'^{-1} g_{\mathcal{H}} \phi(t_i) dg_{\mathcal{H}} \rangle \tag{13}$$

$$= \langle \phi(x'), g_{\mathcal{H}}'^{-1} \int_{\mathcal{G}_{\mathcal{H}}} g_{\mathcal{H}} \phi(t_i) dg_{\mathcal{H}} \rangle \tag{14}$$

$$= \langle g_{\mathcal{H}}' \phi(x'), \int_{\mathcal{G}_{\mathcal{H}}} g_{\mathcal{H}} \phi(t_i) dg_{\mathcal{H}} \rangle \tag{15}$$

$$= \langle \phi(g'x'), \Psi_{\mathcal{H}} \phi(t_i) \rangle \tag{16}$$

$$= l(g'x')_i \ \forall g' \in \mathcal{G} \tag{17}$$

Here, in line 13 we utilize the closure property of a group (since $g_{\mathcal{H}}$ forms a group according to Theorem 2.4). Line 15 utilizes the fact that $g_{\mathcal{H}}$ is unitary, and finally line 16 uses Theorem 2.4. Hence we find that every element of $l(x')$ is invariant to $\mathcal{G}$ observed only through $\mathcal{T}_{\mathcal{G}}$, and thus trivially, $\mathrm{MMIF}(x') = \mathrm{MMIF}(g'x')$ for any $g' \in \mathcal{G}$ observed only through $\mathcal{T}_{\mathcal{G}}$. □