[Reviews · NeurIPS 2017]

Reviewer 1



The paper proposes a new approach for learning representations that are invariant to unitary transformation. I enjoyed reading the paper for its conceptual novelty. The idea is to construct unitary-invariant kernels then use them to construct representations for unlabeled data (the unlabeled data refers to data that are not annotated with the corresponding unitary transformations). There are a few issues the authors should address: 1. What is the integration measure in \Psi_H (as defined on line 199. BTW, please define it clearly as a displayed math)? Why the group (in the RKHS) is locally compact (such that there is a Haar measure)? I think this point worths some thinking over as the compactness relies on the kernel mapping. 2. l(x) is a high-dimensional vector if the number of unlabeled transform data points is large. Thus Line270-271 should be carefully examined as constructing those classifiers become non-trivial. 3. The treatment of unsupervised MMIF (Line270-279) is not very satisfying. How to choose K? What to deal with the randomness of making those binary classifiers? BTW line 270 uses N (# of classes) to represent the number of training samples. 4. Experiments: the paper describes only one experiment where there is only one unitary-group (invariant to pose transformation) being considered. It will be helpful if more experiments are reported. The paper also misses the obvious baseline of not using group-invariant kernels but instead using the same projection step (projecting onto binary SVMs) . 5. Writing: the paper has a significant drop in clarity starting section 3. Other issues should be considered: (1) An example: use an example to illustrate some the concepts such as group, Haar measure, etc (2) Implementation details: many are missing such as how to compute the integration with respect to the Haar measure empirically. Whether the transformation in the experiment study is discrete or not. (3) Keep audience in mind: for example, it is a bit confusing to see Fig3 (and related texts). Since \omega are linear combinations of l(x), so the MMIF would be linear combinations of l(x) too. The extra steps could be explained better. (4) Lack of a discussion section: while most of them are boilerplates, the paper is lacking it and the paper could use the space to discuss some issues and weakness int he proposed approach. What if there are two or more invariant groups? Why MMIF-Pixel is better than MMIF-VGG in the experiments? In general, I like the paper despite above. It is a new way of thinking about representation learning. While a few things still need to be worked out, there are enough sparkling and different thoughts to warrant the acceptance to NIPS to inspire further investigation.

Reviewer 2



#### Paper summary This paper considers the problem of learning invariant features for classification under a special setting of the more general semi-supervised learning; focusing on the face recognition problem. Given a set of unlabeled points (faces), and a set of labeled points, the goal is to learn to construct a feature map which is invariant to nuisance transformations (as observed through the unlabeled set i.e., minor pose variations of the same unlabeled subjects), while being discriminative to the inter-class transformations (as observed through the labeled sample). The challenge in constructing such an invariant feature map is that each face in the labeled set is not associated with an explicit set of possible facial variations. Rather, these minor facial variations (to be invariant to) are in the unlabeled set. The paper addresses the problem by appealing to tools from group theory. Each nuisance transformation is represented as a unitary transformation, and the set of all nuisance transformations is treated as a unitary group (i.e., set of orthogonal matrices). Key in constructing invariant features is the so called "group integration" i.e., integral of all the orthogonal matrices in the group, resulting in a symmetric projection matrix. This matrix is used to project points to the invariant space. The paper gives a number of theoretical results regarding this matrix, and extends it to reproducing kernel Hilbert spaces (RKHSs). The invariant features, termed *max-margin invariant features* (MMIF), of a point x are given by taking invariant inner products (induced by the matrix) in an RKHS between x and a set of unlabeled points. ##### Review summary The paper contributes a number of theorems regarding the group integration, as well as invariance in the RKHS. While parts of these can be derived easily, the end result of MMIFs which are provably invariant to nuisance transformations is theoretically interesting. Some of these results appear to be new. Notations used and writing in the paper are unfortunately unclear in many parts. The use of overloaded notation severely hinders understanding (more comments below). A few important aspects have not been addressed, including why the MMIFs are constructed the way they are, and the assumption that "all" unitary transformations need to be observed (more details below). Computational complexity of the new approach is not given. ##### Major comments/questions I find the MMIFs and the presented theoretical results very interesting. However, there are still some unclear parts which I hope the authors can address. In order of priority, 1. What is the relationship of the new invariant features to the mean embedding features of Raj et al., 2017? Local Group Invariant Representations via Orbit Embeddings Raj et al., 2017 AISTATS 2017 2. The central part of the proposal is the group integration, integrating over "all" orthogonal matrices. There is no mentioning at all how this is computed until the very end on the last page (line 329), from which I infer that the integration is replaced by an empirical average over all observed unlabeled points. When the integration is not over all orthogonal matrices in the group, all the given theorems do not hold. What can we say in this case? Also, how do we know that the observed unlabeled samples are results of some orthogonal transform? Line 160 does mention that "small changes in pose can be modeled as unitary transformations" according to reference [8]. Could you please point out where in [8]? 3. I understand how the proposed approach creates invariance to nuisance transformations. What is unclear to me is the motivation of doing the operations in Eq. (2) and line 284 to create features which are discriminative for classification. Why are features constructed in this way a good choice for being discriminative to between-class transformations? Related, at line 274, the paper mentions about generating random binary label assignments for training. Why random? There is not enough explanation in the text. Could you please explain? 4. The size of the invariant feature map in the end is the same as the number of classes (line 284). However, before obtaining this, an intermediate step in Eq. (2) requires computing a feature vector of size M for each x (input point), where M = number of distinct subjects in the unlabeled set. Presumably M must be very large in practice. What is the runtime complexity of the full algorithm? 5. In Theorem 2.4, the paper claims that the newly constructed transformation g_H is a unitary transformation (preserving inner product) in the RKHS H. However, it appears that only the range of feature map is considered in the proof i.e., the set R := {x | \phi(x) for x in domain of X} and \phi is the feature map associated with the kernel. In general, R is a subset (or equal) of H. The proof of this theorem in the appendix is not complete unless it mentions how to extend the results of the group transformation on R to H. 6. I think Lemma 2.1 requires the group \mathcal{G} to have linear operation. So, not "any" group as stated? ##### Suggestions on writing/notations * \mathcal{X}, \mathcal{X}_\mathcal{G} are overloaded each with 2-3 definitions. See for example lines 119-125. This is very confusing. * Line 280: K appears out of nowhere. I feel that it should be the number of classes, which is denoted by N at line 269. But in the beginning N is used to denote the cardinality of the labeled set... * The paper should state from the beginning that the Haar measure in Lemma 2.1 will be replaced with an empirical measure as observed through the unlabeled sample. It will help the reader. * Line 155: I suggest that the term "generalization" be changed to something else. As mentioned in the paper itself, the term does not have the usual meaning. * There are a few paragraphs that seem to appear out of context. These include line 85, line 274, line 280 and line 284. Why should the paragraph at line 284 be there? I find it confusing. I feel that lines 311-323 should be in the introduction. ------- after rebuttal ------ I read the authors' response. The authors partially addressed the issues that I raised. I have updated the score accordingly.

Reviewer 3



The authors present a novel technique for constructing an SVM invariant to transformations that can either be known a priori or learned from unlabelled data. They provide a solid theoretical foundation for their method and demonstrate it achieving superior performance on the task of face recognition. The paper is overall well written and tells a coherent story from problem statement to theoretical derivation to experimental validation of the method. The problem of designing kernel methods invariant to specific transformations, in particular unknown transformations learned from data, is an important one that should be of interest to a wide audience in the kernel machine learning community. One issue I'd like the authors to clarify is that on line 258 they assume that we have unlabelled data transformed by all actions in the unitary group which usually will not be the case if we try to learn those transformations from an unlabelled dataset. To what extent is this important in practice and what kind of problems could it lead to if the unlabelled dataset didn't include sufficiently many instances of transformed inputs for each data point? A minor problem with notation: line 123 defined X_G as a set {gx,y} while line 125 defines it to be a set {gx}.